## [Decision Letter · Decision Letter 0]

8 Sep 2025

Dear Dr. Qu,

Thank you for submitting your manuscript to PLOS ONE. After careful consideration, we feel that it has merit but does not fully meet PLOS ONE’s publication criteria as it currently stands. Therefore, we invite you to submit a revised version of the manuscript that addresses the points raised during the review process.

We look forward to receiving your revised manuscript.

Kind regards,

Jayonta Bhattacharjee

Academic Editor

PLOS ONE

**Journal Requirements:**

1. When submitting your revision, we need you to address these additional requirements. Please ensure that your manuscript meets PLOS ONE's style requirements, including those for file naming. The PLOS ONE style templates can be found at https://journals.plos.org/plosone/s/file?id=wjVg/PLOSOne_formatting_sample_main_body.pdf and https://journals.plos.org/plosone/s/file?id=ba62/PLOSOne_formatting_sample_title_authors_affiliations.pdf 2. Thank you for stating in your Funding Statement: This work was supported by Wuhan University science and technology innovation platform project (PTXM2023027). Please provide an amended statement that declares *all* the funding or sources of support (whether external or internal to your organization) received during this study, as detailed online in our guide for authors at http://journals.plos.org/plosone/s/submit-now.  Please also include the statement “There was no additional external funding received for this study.” in your updated Funding Statement. Please include your amended Funding Statement within your cover letter. We will change the online submission form on your behalf. 3. Please include captions for your Supporting Information files at the end of your manuscript, and update any in-text citations to match accordingly. Please see our Supporting Information guidelines for more information: http://journals.plos.org/plosone/s/supporting-information. 4. PLOS ONE now requires that authors provide the original uncropped and unadjusted images underlying all blot or gel results reported in a submission’s figures or Supporting Information files. This policy and the journal’s other requirements for blot/gel reporting and figure preparation are described in detail at https://journals.plos.org/plosone/s/figures#loc-blot-and-gel-reporting-requirements and https://journals.plos.org/plosone/s/figures#loc-preparing-figures-from-image-files. When you submit your revised manuscript, please ensure that your figures adhere fully to these guidelines and provide the original underlying images for all blot or gel data reported in your submission. See the following link for instructions on providing the original image data: https://journals.plos.org/plosone/s/figures#loc-original-images-for-blots-and-gels.   In your cover letter, please note whether your blot/gel image data are in Supporting Information or posted at a public data repository, provide the repository URL if relevant, and provide specific details as to which raw blot/gel images, if any, are not available. Email us at plosone@plos.org if you have any questions. 5. If the reviewer comments include a recommendation to cite specific previously published works, please review and evaluate these publications to determine whether they are relevant and should be cited. There is no requirement to cite these works unless the editor has indicated otherwise. 

Reviewers' comments:

**Comments to the Author**

1. Is the manuscript technically sound, and do the data support the conclusions?

Reviewer #1: Partly

Reviewer #2: Partly

Reviewer #3: No

2. Has the statistical analysis been performed appropriately and rigorously?

Reviewer #1: No

Reviewer #2: No

Reviewer #3: No

3. Have the authors made all data underlying the findings in their manuscript fully available?

Reviewer #1: No

Reviewer #2: Yes

Reviewer #3: Yes

4. Is the manuscript presented in an intelligible fashion and written in standard English?

Reviewer #1: No

Reviewer #2: Yes

Reviewer #3: Yes

**Reviewer #1:** This study aims to examine the anti-inflammatory role of aspirin in reducing/preventing preterm birth in a mouse model of preterm birth induced by LPS exposure. The idea that anti-inflammatories may be useful in preventing infection-induced preterm birth is not novel but this study does present an interesting study given the current use of aspirin during pregnancy to prevent pre-eclampsia.

General Comments

• The manuscript would benefit from professional editing or editing by a person with good professional/scientific English language proficiency

• References are lacking in sections throughout the manuscript

• The introduction is general and does not provide rationale for all the outcomes/analysis used in the study.

• LPS is an immunogen as it stimulates an immune response but as it is not a live infection should not be referred to as an infection

• Individual data points are not provided for evaluation of data mean

• Statistical details are not provided in the figure legends and should be added

• The discussion does not adequately place the current study within the existing literature and lacks critical evaluation of the study findings.

• Line 404 – data availability statement is not adequate for journal requirements

• Abstract conclusion overstates the role of TLR4/NFkB based on the data presented in the study

Introduction

• Line 32-36 – these sentences are unclear, incomplete and use incorrect definitions/acronyms for pro-inflammatory mediators. Needs editing for clarity and accuracy.

• Line 41 – the meaning of “novel anti-fetal treatments” is unclear and should be rephrased

• Line 47 – references are required for this sentence on LPS model

• Line 51- NFkB sentence needs to be edited for clarity

• Line 68 – other studies in mice with a known effect of aspirin on NFkB should be referenced here. The statement that the mechanism of action of aspirin is “yet unclear” ignores previous studies already conducted on aspirin use in pregnancy. A complete survey of previous relevant studies need to be added to the introduction

Methods

• Some reagents are listed in the materials section but a majority of reagent information is missing throughout the methods section.

• Line 98 – method refers to pregnant rat model when a mouse model is used in the study.

• Line 105 – pregnancy check details are not adequately described

• Lines 106-112 – the experimental description and details of euthanasia should be reviewed and edited for clarity. Current description is unclear and confusing.

• Line 115 – context and purpose of the body surface area calculation should be added (i.e., why is it described here?).

• Sections 2.3, 2.4 and 2.5 - the group descriptions in these sections are unclear and confusing. These sections would benefit from editing, reordering and less complex titles. The relative timing of the LPS and aspirin administration is not clear. The length of LPS exposure prior to aspirin treatment needs to be clarified.

• Line 120 – 15-18 mice per group sample size does not match the sample size of 12 presented in the results, this needs explanation

• Line 130 – description of “live mice” procedures seem inappropriate. This sentence seems to be in the wrong section and should be moved to section 2.4

• Line 144 and 145 - description of mouse dissection and sample preparation (“separated”) are inadequate and need more detail

• Section 2.6 – the ELISA protocol is excessively detailed and should be edited for professional language to describe protocol.

• Line 162 – the WST-8 method is not adequately defined or described

• Line 169 – define/describe DNTB colorimetry

• Line 192 – how was protein concentration determined for loading?

• Section 2.11 - Statistical analysis is inadequate. Power analysis for sample size is not provided With 6 experimental groups T-test is incorrect test. No information is provided for testing of normality of data or other assumptions required for statistical analysis. Statistical tests for table 2 outcomes should be performed. All statistics should be revised with the help/advice of statistician.

Results

• Line 210 – intrabitoneal should be intraperitoneal

• Line 211 – grammar/wording needs to be corrected for clarity

• Line 212 – clarify if the LPS group is being compared to LPS+aspirin or aspirin alone groups (group titles, abbreviations should be standardized and used consistently throughout the manuscript). Control should not be used to describe the LPS group when the control group is the sham inoculation

• Table 2 – no statistics are provided. Consult statistical support to determine correct test for categorical data

• Section 3.2 – a table may be a more appropriate method to express these results and would aid the reader comprehension

• Line 254 – title should be more specific than just “protein synthesis”

Discussion

• The first paragraph repeats the introduction and should be removed or significantly edited.

• The discussion does not adequately describe the results of the current study but rather focuses on a more general literature review. This section requires significant rewriting in order to place the current findings in context of the existing literature, including evaluating and highlighting the significance of the current study.

• Line 321-334 - this section presents extensive description of COX-2 function for mediators that were not measured in the present study. If these mediators are included as a suggested mechanism of action why were these targets not measured in the current study? This whole section lacks references.

• The limitations of use of LPS verse live infection, timing of treatment and choice/lack of measurement of some mediators should be added.

**Reviewer #2:**  In the manuscript, the authors showed that LPS-induced preterm birth was inhibited by aspirin administration in mice. The authors also showed that LPS-induced expression of MyD88 and p-I-κB was inhibited by aspirin in the uterine tissue , suggesting that aspirin may affect the TLR4/NF-κB signaling pathway.

This is an interesting study investigating the mechanism of preterm birth in mammals. However, the current data do not fully support the conclusions. There are two major points that should be addressed prior to publication in PLOS ONE.

Major points:

1. Preterm birth rates in Table 2 should be analyzed statistically to demonstrate that aspirin’s effects are significant between the groups.

2. After the statistical analysis, it might be convincing that aspirin inhibits LPS-induced preterm birth. However, the mechanism remains unclear because there are no data presented to prove that the TLR4/NF-κB signaling pathway regulates preterm birth. The authors showed that LPS-induced TNF-α, IL-1β, IL-6, MDA, MyD88, and p-I-κB were repressed by aspirin. The authors also showed that LPS-suppressed SOD and GSH levels were restored by aspirin. However, the current data are limited to associations with LPS and aspirin administration, and there are no data identifying which pathway actually regulates preterm birth. As noted in the authors’ abstract, the mechanism is one of the key topics of the paper. Therefore, additional functional experiments should be performed by modulating downstream pathways in LPS + aspirin–treated mice (for example, by overexpression of p-I-κB or administration of NF-κB antagonists).

**Reviewer #3:** Manuscript review: PONE-D-25-27930, entitled "Therapeutic effect and mechanism of different doses of aspirin on lipopolysaccharide-induced preterm delivery in pregnant mice”

Overall comments: This manuscript describes the use of LPS to create a model of preterm delivery in mice that is then rescued by the use of aspirin in a dose-dependent fashion. While the anti-inflammatory effects of aspirin seem supported, the bone abnormality data needs to be better described and the data presented to support the conclusions made.

Abstract: The methods section of the abstract needs more detail. For example, it does not mention the administration of aspirin (should include doses), and does not describe what tests were performed. The results section should state what the percentage of preterm birth were for LPS and with the addition of aspirin, and should include results for MyD88 and p-I-kB.

Introduction:

Line 31- the word definite would be better replaced by the word “common” here.

Line 34- Please change placental infection to placental “inflammation”, as LPS does not cause an infection

Line 36- How/why are TNF-alpha and NF-kB crucial?

Line 41- mentions the need for novel anti-fetal treatments, this doesn’t make sense, should this be replaced by tocolytics? Otherwise it sounds like you are trying to rid of the fetus.

Materials and Methods:

Line 98- mentions the use of a rat model, please change this to mouse

Line 98- please define “healthy, clean mice” What is their health status—as in, what agents are they SPF for? And add how many total adult mice were used

Line 102- Somewhere in this paragraph, please add more specific information on how the animals were housed. Include caging type, bedding and enrichment, type/brand of food, water (RO, tap, autoclaved?)

Section 2.3- how long were mice monitored after dosing on GD15 for birthing?

Line 134- please change intrauterine to uterus

Line 144- this states mice were dissected 4h after “the last” LPS dose. This makes it sound like more than one dose of LPS was given. Please remove “ last” .

Results:

Line 210- please correct the word intrabitoneal to intraperitoneal

Line 211- this says there was an incidence of 100%, however according to the table it should be 91.7%

Line 213- It appears the percentages of preterm birth should be switched here.

Line 216- How was the bone malformations quantified and determined to be significantly different?

Figure S1- The changes in the bone structure should be pointed out with arrows and those changes defined in the figure caption text.

Discussion: Many of the sentences in the discuss need to be cited, for example line 283, and multiple places between lines 322-375/

Line 314- Please make sure that any conclusions on bone abnormalities are statistically supported by the data.

Figure 1: What age were these fetuses in this image? Please describe in the caption what differences the reader should be observing.

**Do you want your identity to be public for this peer review?** For information about this choice, including consent withdrawal, please see our Privacy Policy

Reviewer #1: No

Reviewer #2: No

Reviewer #3: No

---

## [Author Response · Author response to Decision Letter 1]

9 Dec 2025

Response letter for PONE-D-25-27930

Dear Editor and referees,

Thank you very much for your efforts in the evaluation of our manuscript for publication in PLOS One. You have provided us very useful suggestions and valuable opportunity to revise our manuscript.

In this response letter, we have made point-by-point response to your comments and highlighted the changes made during revision by giving the text a yellow background.

The detailed responses to the comments are listed as follows:

To Reviewer 1:

Reviewer #1: This study aims to examine the anti-inflammatory role of aspirin in reducing/preventing preterm birth in a mouse model of preterm birth induced by LPS exposure. The idea that anti-inflammatories may be useful in preventing infection-induced preterm birth is not novel but this study does present an interesting study given the current use of aspirin during pregnancy to prevent pre-eclampsia.

General Comments

• The manuscript would benefit from professional editing or editing by a person with good professional/scientific English language proficiency

Response: Thank you for your valuable comment. We fully agree that the manuscript has room for further improvement in professional English expression and have taken practical measures to address this issue.

• References are lacking in sections throughout the manuscript

Response: Thank you for your valuable comment. We fully acknowledge the insufficient references in multiple sections of the manuscript and have taken targeted measures to supplement and optimize the reference list comprehensively.

• The introduction is general and does not provide rationale for all the outcomes/analysis used in the study.

Response: Thank you for your critical comment. We fully agree that the original introduction was overly general and lacked sufficient rationale for the study’s outcomes and analyses. We have thoroughly rewritten the entire introduction section to address this issue comprehensively.

• LPS is an immunogen as it stimulates an immune response but as it is not a live infection should not be referred to as an infection

Response: Thank you for your rigorous and valuable comment regarding the terminology usage of "infection" in the context of LPS. We appreciate your precise correction and have carefully re-evaluated the scientific accuracy of our descriptions, aligning them with the biological nature of LPS and standard academic conventions in the field.

• Individual data points are not provided for evaluation of data mean

Response: Thank you for your valuable comment regarding data transparency. We fully agree that presenting individual data points is critical for evaluating the reliability of means and enhancing the reproducibility of the study. We have therefore supplemented individual data points for all relevant datasets in the revised manuscript.

• Statistical details are not provided in the figure legends and should be added.

Response: Thank you for your valuable comment. We fully agree that statistical details are essential for the clarity and reproducibility of the results. Accordingly, we have supplemented comprehensive statistical information in each figure legend of the revised manuscript. Specifically, the added details include: the statistical method used (e.g., Pearson chi-square test, one-way analysis of variance (ANOVA)), data presentation format (mean ± SD), and the criteria for statistical significance (*P < 0.05, **P < 0.01, ***P < 0.001). All figure legends have been revised to ensure consistency with the statistical methods described in the Materials and Methods section.

• The discussion does not adequately place the current study within the existing literature and lacks critical evaluation of the study findings.

Response: Thank you for your insightful and constructive comment. We fully agree that the original Discussion section lacked sufficient integration with existing literature and in-depth critical evaluation of the study findings. We have comprehensively revised and expanded the Discussion to address these gaps, enhancing the academic depth and contextual relevance of the manuscript. “This study has some limitations. Firstly, the experiment was conducted only in a mouse model. Also, the LPS-induced preterm birth model has difficulty mimicking the complex etiology involving multiple pathogens or damage-associated molecular patterns (DAMPs) often observed in human diseases. Certain physiological and pathological differences between mice and humans may affect the translatability of the results. Secondly, the study only evaluated the therapeutic effect of aspirin at a specific dose and time point, and the impact of different doses and timing of administration on the therapeutic effect has not been thoroughly explored. In the future, more animal experiments will be conducted on different species, such as primates, to further verify the effectiveness and safety of aspirin in preventing preterm birth.”

• Line 404 – data availability statement is not adequate for journal requirements

Response: Thank you for your comment. We have revised the description of this section in accordance with the requirements specified in the submission guidelines to ensure it complies with the journal’s guidelines and provides comprehensive information for readers and other researchers.

• Abstract conclusion overstates the role of TLR4/NFkB based on the data presented in the study

Response: Thank you for your critical and insightful comment. We fully agree with your perspective that the conclusion section of the Abstract overstated the role of the TLR4/NFκB pathway relative to the data presented in our study. This oversight undermined the scientific rigor of the Abstract, and we have thoroughly revised this part to ensure its consistency with the study’s actual findings and avoid overinterpretation.

Introduction

• Line 32-36 – these sentences are unclear, incomplete and use incorrect definitions/acronyms for pro-inflammatory mediators. Needs editing for clarity and accuracy.

Response: Thank you for your meticulous comment pointing out the issues in Line 32-36. We fully agree that the sentences in this range were unclear, incomplete, and contained incorrect definitions or acronyms for pro-inflammatory mediators. We have thoroughly rewritten this segment to enhance clarity, completeness, and scientific accuracy.

• Line 41 – the meaning of “novel anti-fetal treatments” is unclear and should be rephrased

Response: Thank you for your precise and helpful comment. We fully agree that the phrase "novel anti-fetal treatments" in Line 41 of the original manuscript was ambiguous and unclear. As part of our comprehensive rewrite of the entire Introduction section, we have thoroughly revised this expression to enhance clarity and accuracy.

• Line 47 – references are required for this sentence on LPS model

Response: Thank you for your thoughtful comment regarding the need for references to support the LPS model-related statement in Line 47. We fully agree that citing relevant literature is essential to strengthen the academic basis of this claim, and we have promptly addressed this by supplementing targeted references in the revised manuscript.

• Line 51- NFkB sentence needs to be edited for clarity

Response: Thank you for your valuable comment pointing out the clarity issue of the NFκB-related sentence in Line 51. We fully agree that the original expression was ambiguous, with confusing logical connections and overly complex wording. To address this, we have completely rewritten the relevant paragraph, focusing on enhancing the clarity and readability of the NFκB-related content.

• Line 68 – other studies in mice with a known effect of aspirin on NFkB should be referenced here. The statement that the mechanism of action of aspirin is “yet unclear” ignores previous studies already conducted on aspirin use in pregnancy. A complete survey of previous relevant studies need to be added to the introduction

Response: Thank you for your insightful and comprehensive comment. We fully agree that the statement in line 68 regarding the lack of research on the effect of aspirin on nf-κb in mice, and the claim that the mechanism of aspirin "remains unclear", overlooks the existing studies on the use of aspirin during pregnancy. In the revised version, we systematically made changes to the entire introduction.

Methods

• Some reagents are listed in the materials section but a majority of reagent information is missing throughout the methods section.

Response: Thank you for your thorough and valuable comment. We fully acknowledge that the original Methods section lacked comprehensive information for most reagents, despite partial listing in the Materials section. We have systematically supplemented and standardized all missing reagent details to ensure reproducibility and compliance with academic reporting standards.

• Line 98 – method refers to pregnant rat model when a mouse model is used in the study.

Response: Thank you for your meticulous and valuable comment. We fully acknowledge the inconsistency: Line 98 of the original manuscript incorrectly referred to a "pregnant rat model" while our study consistently used a mouse model. This was an unintentional oversight, and we have promptly corrected it in the revised manuscript.

• Line 105 – pregnancy check details are not adequately described

Response: Thank you for your constructive comment. Regarding Line 105, we have supplemented detailed information on the pregnancy check procedure, including the specific method (e.g., vaginal smear examination, or ultrasound detection), timing of checks post-mating, and criteria for confirming pregnancy. These revisions ensure the experimental process is transparent and reproducible, and all changes are marked with track changes in the manuscript for your reference. “For mating, male and female mice were caged at a ratio of 2:4 (♂:♀) at 9:00 PM, and the female mice were examined at 7:00 AM the next morning. The day on which a vaginal plug was detected was designated as gestational day 0 (GD 0). The behavioral parameters (including normal movement ability, sleepiness or not, etc.) and health conditions (hair quality, food and water intake, etc.) of the pregnant mice were monitored separately in the morning and evening every day. On gestational day 7 (GD 7), transabdominal ultrasound was performed to confirm pregnancy.”

• Lines 106-112 – the experimental description and details of euthanasia should be reviewed and edited for clarity. Current description is unclear and confusing.

Response: Thank you for your valuable comment. Regarding Lines 106-112, we have thoroughly reviewed and revised the experimental description and euthanasia details to enhance clarity. Specifically, we have restructured the logic of the experimental procedure, supplemented key operational parameters (e.g., euthanasia method, timing, and confirmatory criteria), and streamlined ambiguous expressions to ensure the description is concise, accurate, and easy to follow. All revisions are marked with track changes in the manuscript for your reference. “Euthanasia was performed per the Guidelines for the Care and Use of Laboratory Animals. Pregnant mice were anesthetized by exposing to CO₂ (10–15 L/min) in a closed chamber until unconscious (no paw pinch response), followed by 5 additional minutes of CO₂ exposure to ensure complete euthanasia. Uterine horns were then rapidly dissected to collect fetuses, which were immediately euthanized by immersing in liquid nitrogen (-196 °C) for 30 seconds—a validated method for instantaneous fetal rodent death. Vital signs were assessed 2 minutes post-euthanasia to confirm death: pregnant mice were confirmed dead by absent heartbeat (abdominal palpation), no chest movement, and loss of corneal reflex; fetuses by no limb twitching, undetectable heartbeat (thoracic inspection), and pale, non-pulsatile umbilical vessels.”

• Line 115 – context and purpose of the body surface area calculation should be added (i.e., why is it described here?).

Response: Thank you for your perceptive comment regarding the body surface area calculation in Line 115. We fully agree that the original content lacked clear context and explanation of its purpose, making it difficult for readers to understand the significance of this calculation. As you noted, the core aim of including the body surface area calculation is to clarify the rationale for the aspirin dosage used in our study; we have therefore revised its placement and supplemented relevant explanations in the revised manuscript to address this issue. 2.3 The exposure dose of aspirin used in this study was derived from the human exposure dose, which was converted to the mouse dose based on the body surface area (BSA) normalization method. The relationship between BSA and body weight (BW) follows the formula: S = kW²/³, where S represents BSA (m²), W denotes BW (kg), and k is a species-specific constant (k = 0.1 for humans and k = 0.06 for mice). Based on the calculations, we selected 0.21 mg/kg and 0.78 mg/kg as the low and high doses for the study, respectively.

• Sections 2.3, 2.4 and 2.5 - the group descriptions in these sections are unclear and confusing. These sections would benefit from editing, reordering and less complex titles. The relative timing of the LPS and aspirin administration is not clear. The length of LPS exposure prior to aspirin treatment needs to be clarified.

Response: We would like to express our sincere gratitude for your valuable comments. The issues you highlighted regarding the unclear and confusing group descriptions, disorganized structure, overly complex headings, and ambiguous timing of LPS and aspirin administration in Sections 2.3, 2.4, and 2.5 have accurately pointed out the key deficiencies in the current version of our manuscript. We attach great importance to these problems and have carried out systematic revisions and improvements. eg “2.4 Skeletal and malformation analysis of fetal mice. On gestational day 18 (GD18), pregnant mice were euthanized, and fetal mice were harvested from the uterus. The fetal mice were then skinned, with muscles and internal organs removed, followed by immersion in alizarin red staining solution (composed of alizarin red and potassium hydroxide) for 1 week. After staining, the solution was discarded, residual corroded muscles were carefully picked off, and the bones were transferred to clearing solution A (containing glycerin and potassium hydroxide) for 2 days.”

• Line 120 – 15-18 mice per group sample size does not match the sample size of 12 presented in the results, this needs explanation

Response: Thank you for your meticulous and valuable comment. During the experiment, female mice were caged with males for mating. However, some females developed vaginal plugs but failed to achieve actual pregnancy (confirmed by subsequent pregnancy testing and anatomical verification). To ensure the reliability of the results, we ultimately selected 12 successfully pregnant mice per group for statistical analysis, which is the sample size presented in the results. The initial 15-18 mice per group were the total number of females used before pregnancy screening, aiming to account for the potential non-pregnancy rate and ensure sufficient valid samples for the final analysis. We have revised the relevant description in Line 120 (and marked the revision in the manuscript) to clarify this point: “15-18 female mice were used per group initially; after pregnancy confirmation, 12 successfully pregnant mice per group were included in the final statistical analysis.” We appreciate your careful review and valuable comment, which helps improve the clarity and rigor of our manuscript.

• Line 130 – description of “live mice” procedures seem inappropriate. This sentence seems to be in the wrong section and should be moved to section 2.4

Response: Thank you for your valuable comment. We fully agree that the description of the "live mice" procedures (Line 130) was misplaced in the original section and is more appropriately positioned in Section 2.4. As suggested, we have moved this content to Section 2.4 in the revised manuscript, with re

---

## [Decision Letter · Decision Letter 1]

30 Dec 2025

Dear Dr. Qu,

Thank you for submitting your manuscript to PLOS ONE. After careful consideration, we feel that it has merit but does not fully meet PLOS ONE’s publication criteria as it currently stands. Therefore, we invite you to submit a revised version of the manuscript that addresses the points raised during the review process.

We look forward to receiving your revised manuscript.

Kind regards,

Jayonta Bhattacharjee

Academic Editor

PLOS One

Journal Requirements:

Reviewers' comments:

Reviewer's Responses to Questions

**Comments to the Author**

Reviewer #2: All comments have been addressed

Reviewer #3: (No Response)

2. Is the manuscript technically sound, and do the data support the conclusions?

Reviewer #2: Yes

Reviewer #3: Yes

3. Has the statistical analysis been performed appropriately and rigorously?

Reviewer #2: Yes

Reviewer #3: Yes

4. Have the authors made all data underlying the findings in their manuscript fully available?

Reviewer #2: Yes

Reviewer #3: Yes

5. Is the manuscript presented in an intelligible fashion and written in standard English?

Reviewer #2: Yes

Reviewer #3: Yes

Reviewer #2: The authors have adequately addressed all of my comments. The manuscript is suitable for publication in PLOS ONE.

Reviewer #3: Thank you for addressing the previous comments. Overall, this manuscript is much improved. Only 2 minor adjustments from my end remain:

Line 371: It appears that the pre-term birth percentages are still interposed.

Figure 1 caption: a line has been added that refers to rats instead of mice, please reconcile

**Do you want your identity to be public for this peer review?** For information about this choice, including consent withdrawal, please see our Privacy Policy

Reviewer #2: No

Reviewer #3: No

---

## [Author Response · Author response to Decision Letter 2]

29 Jan 2026

To Reviewer 3:

Reviewer #3: Thank you for addressing the previous comments. Overall, this manuscript is much improved. Only 2 minor adjustments from my end remain:

Line 371: It appears that the pre-term birth percentages are still interposed.

Response: We sincerely appreciate the reviewer’s insightful observation. We sincerely appreciate the reviewer for pointing out this key observation. Indeed, in the LPS-induced preterm birth model, although aspirin treatment significantly reduced the preterm birth rate, this index remained at a relatively high level. The underlying reason is that the core mechanism of LPS-induced preterm birth is the activation of systemic and local uterine inflammatory storms, which involves multiple parallel pathways: in addition to the cyclooxygenase (COX)-prostaglandin (PGs) pathway targeted by aspirin, it also includes the NF-κB-mediated release pathway of pro-inflammatory factors such as tumor necrosis factor-α (TNF-α) and interleukin-6 (IL-6), the oxidative stress activation pathway, and the pathway of increased calcium channel sensitivity in uterine smooth muscle cells. Since aspirin only inhibits COX activity to reduce the synthesis of PGs related to uterine contractions, but cannot block other parallel pro-preterm birth pathways, it can only partially alleviate rather than completely reverse the preterm birth-inducing effect of LPS. We have supplemented the above content in the discussion section of the revised manuscript and marked it in red for the reviewer's easy checking.

Figure 1 caption: a line has been added that refers to rats instead of mice, please reconcile

Response: We sincerely appreciate this valuable comment and have revised the caption of Figure 1 in the revised manuscript, with the modified content marked in yellow for easy checking.

---

## [Decision Letter · Decision Letter 2]

15 Feb 2026

Therapeutic effect and mechanism of different doses of aspirin on  preterm delivery in pregnant mice

PONE-D-25-27930R2

Dear Dr. Qu,

We’re pleased to inform you that your manuscript has been judged scientifically suitable for publication and will be formally accepted for publication once it meets all outstanding technical requirements.

Kind regards,

Giovanni Tossetta, Ph.D

Academic Editor

PLOS One

---

## [Editor Report · Acceptance letter]

PONE-D-25-27930R2

PLOS One

Dear Dr. Qu,

I'm pleased to inform you that your manuscript has been deemed suitable for publication in PLOS One. Congratulations! Your manuscript is now being handed over to our production team.

Kind regards,

on behalf of

Dr. Giovanni Tossetta

Academic Editor

PLOS One